# Software Architecture Patterns for Extending Sensing Capabilities and Data Formatting in Mobile Sensing

**DOI:** 10.3390/s22072813

**Published:** 2022-04-06

**Authors:** Jakob E. Bardram

**Affiliations:** Department of Health Technology, Technical University of Denmark, DK-2800 Kongens Lyngby, Denmark; jakba@dtu.dk

**Keywords:** mobile computing, mobile sensing, wearable sensing, mobile health, mHealth, digital phenotyping, Open mHealth, electrocardiography, ECG

## Abstract

Mobile sensing—that is, the ability to unobtrusively collect sensor data from built-in phone and attached wearable sensors—have proven to be a powerful approach to understanding the behavior, well-being, and health of people in their everyday life. Different platforms for mobile sensing have been presented and significant knowledge on how to facilitate mobile sensing has been accumulated. However, most existing mobile sensing platforms only support a fixed set of mobile phone and wearable sensors which are ‘built into’ the platform’s generic ‘study app’. This creates some fundamental challenges for the creation and approval of application-specific mobile sensing studies, since there is little support for adapting the sensing capabilities to what is needed for a specific study. Moreover, most existing platforms use their own proprietary data formats and there is no standardization in how data are collected and in what formats. This poses some fundamental challenges to realizing the vision of using mobile sensing in health applications, since mobile sensing data collected across different phones and studies cannot be compared, thus hampering generalizability and reproducibility across studies. This paper presents two software architecture patterns enabling (i) dynamic extension of mobile sensing to incorporate new sensing capabilities, such as collecting data from a wearable sensor, and (ii) handling real-time transformation of data into standardized data formats. These software patterns are derived from our work on CARP Mobile Sensing (CAMS), which is a cross-platform (Android/iOS) software architecture providing a reactive and unified programming model that emphasizes extensibility. This paper shows how the framework uses the two software architecture patterns to add sampling support for an electrocardiography (ECG) device and support data transformation into the new Open mHealth (OMH) data format. The paper also presents data from a small study, demonstrating the robustness and feasibility of using CAMS for data collection and transformation in mobile sensing.

## 1. Introduction

Mobile sensing enables unobtrusive collection of sensor data from build-in phone sensors and attached wearable sensors. It has been shown that indicators of behavioral, social, psychological, and health status can be derived by collecting continuous and real-world data and applying advanced algorithms to it [1]. A wide range of research studies have applied mobile sensing to health and wellness applications [2], including, for example, the EmotionSense [3], BeWell [4], and StudentLife [5] systems that classify physical activity, sleep, and social interaction based on sensor data. Similarly, a number of mobile health (mHealth) applications for mental health have been proposed [6], and studies have demonstrated correlations and predictive power between phone-based features on physical activity, mobility, social activity, phone usage, and voice data on the one hand, and mental health symptoms in, for example, depression [7], bipolar disorder [8,9], and schizophrenia [10] on the other.

To support easy configuration and the deployment of mobile sensing studies, a number of mobile sensing platforms and programming frameworks have been proposed. These aim at providing more general-purpose support for mobile sensing, including support for the configuration of sampling protocols, accessing low-level sensor data, and handling and storing these data (see Kumar et al. [11] for a review). Most of this research has focused on providing easy-to-use *platforms* for the collection of data from mobile phones and storing this in a cloud-based infrastructure. These platforms typically have the options to configure a sampling protocol (or ‘study’), enroll a set of participants, deploy the study onto the participants’ mobile phones and automatically collect data in a cloud infrastructure, which can be accessed from a web portal. Contemporary examples of this approach include Purple Robot [12], Sensus [13], the AWARE Framework [14], the Beiwe Research Platform [15], mCerebrum [16], RADAR-base [17] and LAMP [18], which all are quite elaborate and mature platforms. These platforms are designed for research and target experimental behavioral researchers as end-users; the goal is to allow researchers to easily configure a study, enroll participants, deploy the study on the participants’ phones, and collect the data automatically with as little interaction with participants as possible.

In our research we have, however, identified three other types of requirements for mobile sensing that such platforms would need to support. First, often there is a need for designing a custom app for a particular domain and patient group, and to be able to add support for mobile and wearable sensing to such special-purpose apps. Often the motivation for participants to engage in these studies relies on that “there’s something in it for them” [19], which again means that the app should not be designed with the researcher in mind, but the participant. Most of the mobile sensing platforms provide a ‘standard’ app for the participants, with limited support for customization. Rather than such a standard app, there is a need to have a mobile and wearable sensing *programming framework* that allows researchers to easily add *relevant* data sampling to a special-purpose app during design and implementation. Second, there is an increasing focus on privacy in the collection of data from users. The European Union (EU) General Data Protection Regulation (GDPR) stipulates that only data which are strictly needed for a study must be collected. The recent strict privacy policies of the App Stores (Apple and Google) state that they do not approve and publish apps that make use of extensive passive sensing, if data are not used in the app and shown to the user. For example, if your app is not using and showing location to the user, the app is not allowed to collect location data. Most of the existing platform apps collect a fixed set of data types, and even if the collection of a data type can be disabled on runtime in a study configuration, the app still declares that it is collecting this type of data. Therefore, there is a need to be able to *flexibly* add and remove sensing of different data types from an application-specific app in order to make it legal according to GDPR and to get it approved in the App Store. Third, there is a need for standardization in mobile sensing. This includes both how a study protocol is defined, how data are collected, and not least the format of the data collected. Currently, each mobile sensing platform uses its own proprietary study configuration and data format. This hampers cross-study comparison, replicability, and the collection of large compatible datasets for analysis. Moreover, in the medical domain, a wide range of standards are available and the collection of mobile sensing data in the health domain need to support such medical data standards. Therefore, there is a need to support different data formats in mobile sensing.

This paper presents two software architecture patterns [20] that address these three requirements by showing how a programming framework for mobile sensing can support dynamic extensibility of data sensing capabilities and for data transformation and formatting. These two design patterns are named ‘Sampling Package’ and ‘Data Transformer’ and are presented in Section 2 and Section 3, respectively. The presentation of the patterns loosely follows the Design Pattern book [21] by stating the pattern’s name, classification, intent, motivation, applicability, structure, dynamic behavior, and consequences. The patterns have been derived from, and implemented in, the CARP Mobile Sensing (CAMS) programming framework [22] CARP Mobile Sensing (CAMS) is part of the CACHET Research Platform (CARP) platform [23], where CAMS is the mobile sensing components of CARP. CAMS is implemented in Flutter using the Dart programming language, and the examples provided in Section 2 and Section 3 are therefore written in Dart. By using the ‘Sampling Package’ and ‘Data Transformer’ patterns, CAMS becomes highly extensible in a number of ways: it allows for implementing new data sampling methods (including both phone sensing, external wearable devices, and cloud-based services); it supports the creation of new data transformers (both for privacy reasons and data standards), and it allows for creating custom data managers, which can upload data in a specific format to specific servers, or other kinds of data off-loading. The patterns are exemplified by showing how they are used to extend CAMS to collect data from an electrocardiography (ECG) device and to support data in the Open mHealth (OMH) data format [24,25]. The paper also demonstrates the feasibility of the sampling package and data transformation patterns by reporting from a study where mobile sensing data are collected via different sampling packages and transformed into the OMH data format before storage.

## 2. The ‘Sampling Package’ Software Architecture Pattern

The ‘Sampling Package’ pattern can be used to specify what data to be collected, the format of this data, and how data are acquired, including the use of devices or sensors. A sampling package typically handles a collection of related measure types. For example, a ‘connectivity’ sampling package can collect data on connectivity status, wifi, and bluetooth. The ‘Sampling Package’ pattern is both a structural and behavioral pattern, which defines the structure of the packages and its associated classes plus its dynamic behavior when used in mobile sensing.

### 2.1. Motivation and Application

Different mHealth applications using mobile sensing will need to collect different types of data from different devices, and these data types are not known to the mobile sensing framework. Therefore, there is a need for supporting the future addition of collecting data. Moreover, an mHealth app which is to be released in an app store (e.g., Apple App Store or Android Play) needs to declare what data it is collecting, such as location data. However, if the collection of location data were ‘built into’ the mobile sensing framework, then all apps would need to declare that they were collecting location—even though they were not. With the current strict privacy rules in the app stores, this would lead the app to be rejected. Therefore, an app—and hence the sensing framework—needs to include and hence declare *only* those data types which are actually used.

The ‘Sampling Package’ pattern solves three problems; (i) encapsulation of the collection of different but related data types; (ii) specification of data types and how they are collected, e.g., using a dedicated device; and (iii) dynamic loading and use of relevant data sampling specified in the study protocol.

### 2.2. Structure

The sampling package unified modelling language (UML) class diagram is illustrated in Figure 1. A sampling package implements the following classes:SamplingPackage—the overall specification of the packages, specifying which measures it can collect (dataTypes), what device type it supports, the device manager for its device, and the list of operating system (OS) permissions needed for this package.DeviceManager—specifies how a device is managed. Note that a device may be the smartphone itself or an external device, such as an ECG sensor. Each sampling package only handles one device type but several probes can use the same device.Probe—the concrete implementation of the collection of data of a specific type from the underlying OS, connected devices, or external services. Data are streamed from the probe as DataPoints in the data stream.DeviceConfiguration—configuration of the device for this study.Data—specifies the format of the data collected and the data itself.

Figure 1 also illustrates how these classes are implemented (specialized) as part of the MovisensSamplingPackage The Movisens EcgMove4 is a wearable movement and ECG monitor. The Movisens package can collect data of the type ‘dk.cachet.carp.movisens’, supports the ‘dk...MovisensDevice’ device type via the MovisensDeviceManager, and can create a MovisensProbe, which uses the Movisens Android libraries to collect Movisens data objects, including heart rate (HR) data.

Figure 2 illustrates the dynamic behavior of the sampling package pattern. First, the sampling package is registered in the SamplingPackageRegistry, which in turn will register the package’s device manager in the DeviceRegistry. The StudProtocol contains a specification of what measures to collect on and what devices to use (see Section 4.1 for details on how a study protocol is defined). A StudyExecutor is responsible for executing a study as defined in the study protocol. The study executor is configured from the study protocol, which in turn configures the device manager for each device specified in the protocol. When the executor is initialized, it iterated through the list of measures and uses the SamplingPackageRegistry to lookup and create probes for each data type that a measure specifies. Each probe is initialized with its measure. When the executor is resumed, each of its probes is resumed, which looks up the device in the DeviceRegistry, which then is used to collect the data points. Each data point is added to the data stream of the probe, and all data streams from all probes are aggregated in the executor.

### 2.3. Example

In the following, we shall use the Movisens sampling package as an example of a sampling package that defines how to collect data from the Movisens ECG device. Figure 10 shows the most important part of the Movisens sampling package implementation. The package defines what data types it supports (in this case only the MOVISENS data type), the device type, and what OS-level permissions it needs (these details are not listed, but include access to Bluetooth). The onRegister callback implementation is presented later, and the create method creates a new MovisensProbe, which is shown in Figure 11.

**Listing 1 sensors-22-02813-f010:**
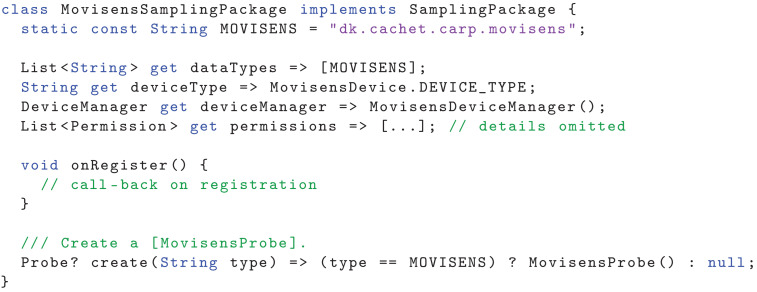
Definition of the Movisens sampling package.

The MovisensProbe is implemented using a StreamProbe, which can handle data coming from a stream of data points. The StreamProbe handles the life cycle of a probe (i.e., initialize, resumed, pause, restart, and stop) and the MovisensProbe merely has to provide the stream of data as Data objects. This is done by listening on the stream of Movisens data coming from the Movisens device manager (accessed as deviceManager.movisens. movisensStream), and map the incoming events to the MovisensData data format. The MovisensDeviceManager implements the connection to the Movisens device using a Movisens Flutter plugin. The Movisens device manager is configured using a MovisensDevice Configuration, which is defined in the study protocol. This device configuration specifies details of the Movisens device, including its Bluetooth Low Energy (BTLE) address, name, location, and details on the user such as age, height, and weight (used for the calculation of metabolic level). The Movisens device manager implements the attributes and methods shown in Figure 1, including status, and the life-cycle methods onConfigure(), canConnect(), onConnect(), and onDisconnect(). These call-back methods uses the Movisens Flutter plugin to configure, connect, and disconnect to/from the Movisens device.

**Listing 2 sensors-22-02813-f011:**
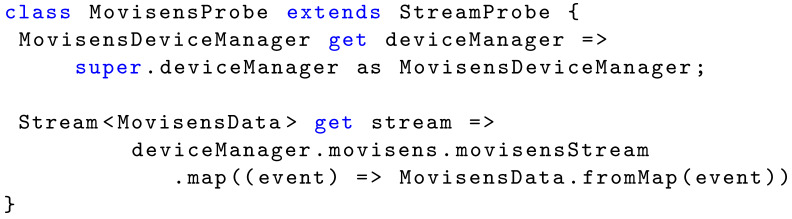
Definition of the Movisens probe.

### 2.4. Consequences

The sampling package pattern allows the design of a mobile sensing framework that is born ‘empty’, i.e., a framework that does not support any sensing probes, devices, or data types. The application programmer can then include only those sampling packages and hence data types which are needed for a specific app. For example, as further presented below, CAMS only have two built-in sampling packages (the sensor and device packages). The remaining packages are external to CAMS and are only included and linked if an application needs them. This is illustrated in Section 4.2, where the included packages are shown in green whereas the external packages are shown in purple. This modular design of CAMS has a number of benefits, including decreasing app size, reducing dependencies to only those packages needed, not having to ask the user for permission to access data sources that are not used, and—not least—that the app developer can reduce the privacy permissions during approval of the app in the App Stores. The sampling package concept also provides a strong modularization model for adding *external* devices to an app. As illustrated above, the real-time ECG data collection from the Movisens EcgMove4 device is implemented as a separate sampling package, which encapsulates the low-level details of handling this device and its data formats. Similarly, support for collecting sensor data from the eSense wearable computing platform [26] have been implemented as a separate sampling package which can collect the button press, accelerometer, and gyroscope sensor events from the eSense device [27].

## 3. The ‘Data Transformer’ Software Architecture Pattern

The ‘Data Transformer’ pattern can be used to specify how data are transformed from one format to another. This includes transformation from the format data are collected in, to standard data formats for mHealth applications such as the OMH or Fast Healthcare Interoperability Resources (FHIR) formats. However, transformation can also support privacy by obfuscating or encrypting data. Data transformation can be linked to a sampling package, which can provide so-called ‘transformers’ for the data types it collects. For example, a ‘communication’ sampling package which can collect data on SMS messages, calendar entries, and phone calls can provide a set of default ‘privacy’ transformers which removes sensitive data from the collected communication data, such as message content, phone numbers, attendees in calendar entries, and so forth. Data transformation can be linked so that several transformers can be chained. In this way, data can first be obfuscated to preserve privacy and then transformed to the OMH format. The ‘Data Transformer’ pattern is mainly a behavioral pattern, which defines the dynamic behavior of data transformation across several transformers when used in mobile sensing.

### 3.1. Motivation and Application

Data transformation addresses several issues in mobile sensing. First, different mHealth applications using the data collected by mobile sensing might need to use different data formats, both locally in the app and when storing this data. This also include different data storage solutions—such as cloud-based infrastructures—which expects data to be uploaded in a specific format. For example, if you want to store data in the Open mHealth Storage Endpoint, you need to upload data in the OMH format. Second, in order to preserve users’ privacy at the source, i.e., on the phone, local data transformation in terms of obfuscation and encryption might be needed. Third, an application may need to enrich the collected data with additional, supplementary data which is not part of the data collection supported in the mobile sensing framework. This can be achieved by transforming the data by adding additional information. For example, by adding location information as a geo-tag to a survey, thereby recording where the user filled in the survey.

### 3.2. Structure

The central part of the Data Transformer pattern is the DataTransformer function that can transform one piece of *Data* into another. Figure 12 shows the type definition of the DataTransformer function.

**Listing 3 sensors-22-02813-f012:**
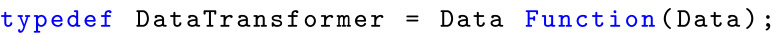
The type definition of the DataTransformer function.

The UML class diagram for the Data Transformer pattern is illustrated in Figure 3, showing the following classes:DataTransformerSchema—a data transformer schema contains a set of data transformers, which can transform data within a specific name space. Specific schemes for specific name spaces inherent from the DataTransformerSchema, such as an OMH, FHIR, and privacy schemes;TransformerSchemaRegistry—a transformer schema registry holds a set of data transformer schemes, such as the ones shown in the class diagram;DataTransformerFactory—a data transformer factory knows how to create a transformer. A transformer factory can be any function anywhere in the code. However, as illustrated in Figure 3, it is common that the ‘new’ data class itself implements its own factory, thus knowing how to transform data. For example, the OMHHeartRateDataPoint factory method transformer implements the transformation from CARP HR data to OMH HR data points.

Figure 3 also illustrates how any ‘new’ data item that a transformer can produce inherits (is a) Data class itself. This applies for the OMHHeartRateDataPoint, OMHStepCount DataPoint, FHIRHeartRateObservation, and all other ‘new’ data types. This implies that a newly transformed data object can be feed to another transformer and further transformed, thereby enabling a chain of transformers (similar to the pipes-and-filter software pattern [20]).

The dynamic behavior of the data transformer pattern can be divided into two phases: (i) setup and (ii) transformation. Figure 4 illustrates the setup phase. Setup is typically performed in the sampling package upon registration. For each name space, a transformer schema is looked up in the transformer schema registry, and a transformer for relevant data types are added to each schema. Figure 4 shows an example from the Movisens sampling package, which adds three transformers: (i) one that can transform CARP HR data into OMH HR data points, (ii) one that can transform CARP pedometer data into OMH step count data points, and (iii) one that can transform CARP HR data into FHIR HR observations.

Figure 5 illustrates the transformation phase. Transformation can be performed anywhere in an app, but if the Data Transformation pattern is combined with the Sampling Package pattern then transformation can be performed on the data stream of data produced in the StudyExecutor (see Figure 2). The study executor looks up a transformer schema for a specific name space (e.g., FHIR) and asks this schema to transform any data object, which comes from the data stream (which again comes from the probes). When the transformer schema is asked to transform a piece of data, it checks if it has a transformer (added during the setup phase). If so, it uses this transformer to transform the data, otherwise it does nothing (i.e., returns the data unchanged).

### 3.3. Example

Figure 3, Figure 4 and Figure 5 show how the ‘Data Transformer’ pattern is used in the Movisens sampling package. Figure 13 shows how the setup phase of the patterns is implemented as part of the onRegister() method of the Movisens sampling package. Here the three transformers are added to the OMH and FHIR transformer schemes.

**Listing 4 sensors-22-02813-f013:**
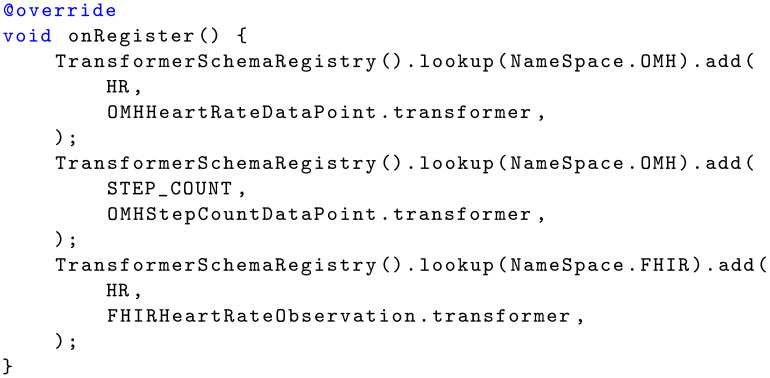
Setting up DataTransformer functions as part of registering the Movisens sampling package.

Transformation can now be performed as shown in Figure 14 where CARP HR data are transformed to an OMH heart-rate data point.

**Listing 5 sensors-22-02813-f014:**
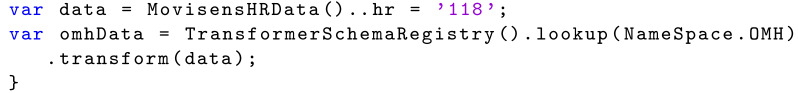
Lookup and use of a DataTransformer function.

Transformers can also be used in a stream, as shown in Figure 15 where all CARP data from the dataStream are transformed to OMH data points.

**Listing 6 sensors-22-02813-f015:**
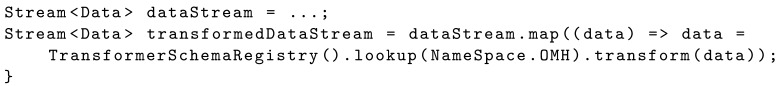
Using a transformer in a stream of data.

Finally, transformers can also be chained in streams, as shown in Figure 16. Here all CARP data from the dataStream are first privacy protected using the "privacySchemaName" transformer, and then transformed to OMH data points. Note that only data which has a CARP-to-OMH transformer is transformed in these streams. If no transformer exists, the original CARP data item is kept unchanged. We will see an example of this in the study reported in Section 5.

**Listing 7 sensors-22-02813-f016:**
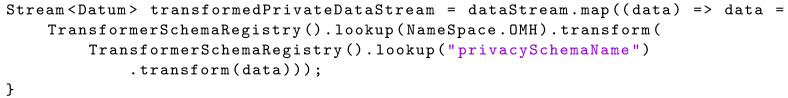
Chaining two transformers in a stream of data.

### 3.4. Consequences

The data transformer pattern allows mobile sensing such as CAMS to implement on-board data transformation, which can be used for local data privacy protection or transformation before storage or upload. This allows a mobile sensing framework to be used in many different types of mHealth applications and using many different types of data backends. Hence, a sensing framework such as CAMS is not tied to a specific infrastructure such as CARP and its data format, but can be extended to work with any data infrastructure and data formats.

One of the limitations to the data transformer pattern is that it implements a one-to-one transformation where one CARP Data object is transformed into exactly one other Data object. There is no support for aggregation or splitting of Data objects. This could be a topic for the extension of the pattern or by embedding this one-to-one mapping as part of a more general pipes-and-filter software architecture.

## 4. Enabling Extensibility and Data Transformation in the CAMS Software Architecture

The ‘Sampling Package’ and ‘Data Transformer’ design patterns have been derived from, and used in, the CAMS software architecture and help to achieve the non-functional software architecture goal of being highly extensible. CAMS has been evolving over eight major releases, and has been used in the design and implementation of several released mHealth applications targeting mental health [28], cardiovascular disease (CVDs) [29], and diabetes [30]. These applications are very different in their design and requirements for mobile sensing, and hence testifies to the wide applicability of CAMS for different types of applications. This section outlines how the patterns come into play in CAMS and focuses on its extensibility. For a more detailed presentation of CAMS and its programming application programming interface (API), runtime model, and performance, please see the CAMS technical report [22] and the online resources listed in Appendix A.

### 4.1. Study Protocol

Data collection is configured in a StudyProtocol, which is shown in Figure 6. A StudyProtocol holds a set of Triggers, which can trigger one or more Tasks, which again hold a set of Measures. A Measure specifies which type of data should be collected. In short; triggers defined *when* to collect data, tasks defined *which* measures to collect, and the measures defined *what* data to collect. Using the Device class, the study protocol also specifies which devices should be used for data collection. Note that a protocol has no technical dependencies on any particular devices, sensor technology, or services. By specifying a set of generic Devices and Measures it in an abstract manner describes why, when, and what data should be collected. It is up to the runtime infrastructure of the sensing framework to interpret and execute the protocol in a ‘best-effort’ manner, using the sensors and devices available on a specific phone. Figure 17 illustrates how a study protocol is configured in Dart in CAMS. First (line 2–5) the study protocol is created and then a SmartPhone device is added to the protocol (line 8–9). Then the measures are configured consisting of a delayed trigger (DelayedTrigger) that triggers one task (AutomaticTask), which automatically collects accelerometer and gyroscope data. Note that the accelerometer and gyroscope measures are defined in the sensor sampling package (line 16–17).

**Listing 8 sensors-22-02813-f017:**
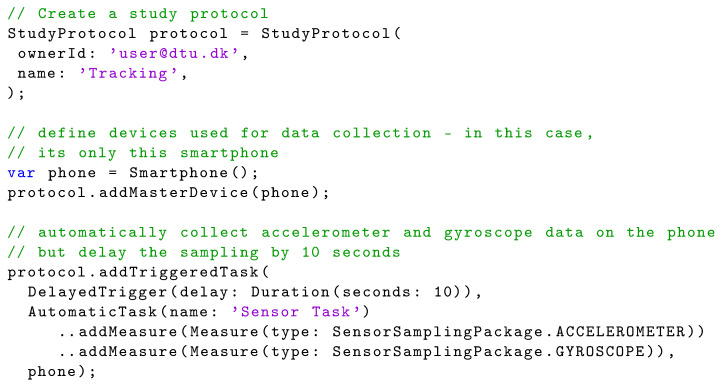
Basic Dart code for setting up a study protocol in CAMS.

### 4.2. Runtime Architecture

Figure 7 shows the overall layered software architecture of CAMS, and illustrates how the ‘Sampling Package’ and ‘Data Transformation’ design patterns are central to this architecture. The CAMS runtime model consists of three main layers (from the bottom):The **Sampling Packages** layer, which can collect a specific type of measure. CAMS comes with two built-in sampling packages (the DevicePackage and SensorPackage shown in green). But by using the ‘Sampling Package’ pattern, most sampling packages are available as external packages which are used in the app, as needed (shown in purple). In this way, an app only needs permissions to access sensors, which are needed in a specific app. The ContextPackage is an example of an externally loaded sampling package that uses location permissions.The **Client Manager** layer, which runs many of the processes and registries from the two patterns. The StudyController holds the StudyExecutor, one or more DataTransformers, the DataManager, and the PermissionManager. The Study- Controller is configured by adding a study protocol to the SmartPhoneClientManager. Once configured, the StudyExecutor is responsible for executing the study deployment, i.e., collecting the data following the pattern shown in Figure 2. As also shown in Figure 2, Figure 4 and Figure 5 the StudyExecutor uses a set of registries to dynamically look the sampling packages, data transformers, and sensing probes that are needed for executing the study protocol. A study protocol also specifies how to store data by specifying a ‘data endpoint’, which again is used to look up an appropriate DataManager in the DataManagerRegistry. This is, howerver, outside the two design patterns discussed in this paper.The **Service** (top) layer holds one or more DeploymentServices, which is able to manage and deploy the StudyProtocol. Figure 7 illustrates two such services; the SmartphoneDeploymentService that can deploy study protocols locally on the phone and the CarpDeploymentService that can download study protocols from the CARP Web Services (CAWS). The service layer also holds the DataManager services, which know how to store, save, or upload data. Figure 7 illustrates two such data managers; the FileDataManager which stores data locally in files on the phone and the CarpDataManager which uploads data to CAWS.

Beneath (and outside of) CAMS, each sampling package uses one or more Flutter plugins to access sensors, processes, data, and services in the underlying OS, connected wearable devices, or online services. For example, accessing the phone’s onboard pedometer, connecting to the Movisens ECG monitor via BTLE, or accessing weather information via a web API. As illustrated in Figure 7, the CAMS runtime make extensive use of registries in which different components can be registered and retrieved at runtime. These registries are core to the extensibility of the framework since they allow for adapting and extending how data are acquired, formatted, anonymized, stored, and uploaded.

Figure 18 shows how the CAMS sensing runtime is configured and started, and how the sensed data can be used in an app. Deployment of a study protocol on a device requires that the device knows the id of the study protocol, and its own role in the study (line 2–3). As shown in Figure 6, the role of a device in the study protocol is specified as part of the device configuration of the protocol (called the roleName). The study id and role are typically obtained via the user—either by typing it in or downloading a configuration where this is specified, e.g., in a QR code. Then (line 6–7) a client manager is created and configured, which in turn initializes and configures all the client manager components shown in Figure 7. Once the client manager is configured, a study controller is created (line 10–11) by specifying the deployment id and the role name of the device (in this case the phone), after which the controller is ready for executing the study locally on the phone. The controller can be configured, if needed (line 14) and sensing is resumed/started (line 15). The collected data are now available in the data stream, which can be used in the app (line 20). At any time, sensing can be controlled via the controller’s life cycle methods; resume, pause, restart, and stop (e.g., line 25).

**Listing 9 sensors-22-02813-f018:**
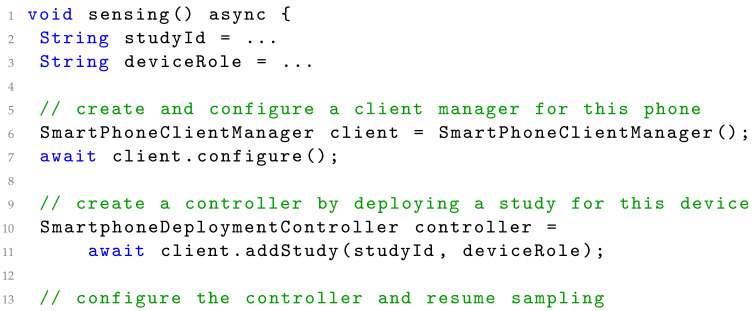
Basic Dart code for configuring, starting, and using mobile sensing in CAMS.

### 4.3. Implementation and Availability

CAMS is implemented in Flutter using the Dart language, and is available as open-source under an MIT license. Flutter is a cross-platform toolkit for building natively-compiled applications for mobile, web, and desktop from a single codebase [31]. Dart is a modern object-oriented, reactive programming language optimized for non-blocking user interface (UI) programming with a mature and complete async-await event-driven code style, paired with isolate-based concurrency model. The implementation of CAMS particularly exploits three core aspects of Dart: (i) the asynchronous non-blocking programming style using the Future construct, (ii) the event-driven reactive stream model using the Stream API, and (iii) access to native OS processes via the PlatformChannel API.

Note that the Flutter plugins shown in Figure 7 are not part of CAMS, but are 3rd party plugins available from the Flutter package repository (https://pub.dev/ (accessed on 15 February 2022)). The CARP team contributes to this library of Flutter plugins. However, these plugins are designed to be general-purpose and are not specific to CAMS. CAMS-specific use of these plugins is implemented in the different sampling packages.

Sampling packages are Flutter packages, which are also released on pub.dev. This has three important implications: First, sampling packages can be downloaded and added to a Flutter app as needed when the app is build. Second, in contrast to most other sensing frameworks which has all the probes built-in, an app developer only needs to download and add sampling packages which are needed for his/her specific app. Hence, if context information is not needed for an app, the context package and its permission to collect, e.g., location is not linked and used. Third, application programmers can share CAMS sampling packages with each other via pub.dev, which is the official Flutter/Dart code sharing infrastructure, which also ensures continuous quality assessment of the code.

CAMS has been designed, implemented, and tested over the course of eight major releases and the core framework and all of its associated plugins, sampling packages, and data backends have been released on the Flutter pub.dev software package repository. Appendix A provides an overview of all the online CAMS resources, including API documentation and online tutorials, and Appendix B contains an overview of currently available sampling packages and their measures.

## 5. Study

In order to demonstrate the stability of the data sampling packages and data transformation in CAMS, we report from a small mobile sensing study. The purpose of this study is to verify and evaluate the usefulness of sampling package and data transformation patterns as implemented in CAMS. Hence, focus is on demonstrating how a wide range of measures can be collected across several sampling packages, and that data from these sampling packages can correctly be transformed from one data format to another. In this study, we used the context sampling package for the collection of a range of contextual data points, including location, activity, air quality, and weather. In addition to these contextual data types, we also collected data using the device and sensor packages, including battery, memory, light, noise, screen on/off, and step count. The study protocol was configured to transform collected data points into the org.openmhealth data format, if possible. The detailed CAMS study protocol definition is included in Appendix C. Data were collected over a period of two days (50 h) from one person using a Samsung Galaxy S10e, Android OS level 11. Table 1 shows the total number of collected data points distributed on measure type.

Figure 8 shows the number of collected data types on an hourly basis over the 50 h period. Figure 9 shows the distribution of the total number of collected data. We can observe that 11 different types of data were collected with a wide variety over the 50 h period, reflecting differences in activity level and circadian rhythms. Of these 11 types of data, OMH only supports location (named geoposition) and activity (named psychical-activity), but we see that the collected data were correctly transformed into these two types. Of all data, 42% are of type geoposition and 6% are of type psychical-activity. The rest of the data are kept in the default dk.cachet.carp name space.

## 6. Conclusions

This paper has presented the ‘Data Sampling Package’ and ‘Data Transformation’ software patterns. Data sampling packages enable the application programmer to implement support for collecting new data types and plug this into a mobile sensing framework. Sampling packages are self-contained packages that can be added to a mobile sensing framework and hence an app as needed. This allows the programmer to include only those sensing capabilities which are needed in an application-specific app, thereby reducing app size and complexity, and adhering to the privacy rules of, e.g., GDPR and the App Stores. Data transformers allow the application programmer to transform the data collected by a sensing framework into any other format needed either locally in the app, or when storing or uploading data. This also enables the privacy protection of data via obfuscation or encryption, and transformers can be chained for multiple transformation. The design patterns presented in this paper are generic in nature and can be used for ensuring a high degree of extensibility when designing a mobile sensing framework.

The utility of these design patterns has been shown by implementation in the CARP Mobile Sensing (CAMS) programming framework. CAMS is a cross-platform (Android/iOS) mobile programming framework, which in addition to state-of-the-art mobile and wearable sensing provides a modern reactive programming API with a unified approach to data sampling, management, transformation, usage, storage, and upload across different types of data sources and data storage facilities. CAMS provides support for data sampling from on-board mobile sensors (e.g. accelerometer, location, and step counter), from phone logs (e.g. call log), from off-board wearable sensors (e.g. ECG monitor), and web-based services (e.g. weather forecast). In CAMS, sampling packages are implemented as Flutter packages and can be downloaded and uploaded to the Flutter package sharing repository pub.dev. This fosters visibility, availability, and quality assurance of the sampling packages.

CAMS has been used in the design and implementation of several mHealth applications targeting mental health [28], CVDs [29], and diabetes [30]. From an application and UI point-of-view, these applications are quite different both in terms of technical design and in data collection and management, which hence demonstrate the flexibility and extensibility of CAMS for mobile and wearable sensing for a broad range of applications. We hope that others also can benefit from using and extending CAMS.

## Figures and Tables

**Figure 1 sensors-22-02813-f001:**
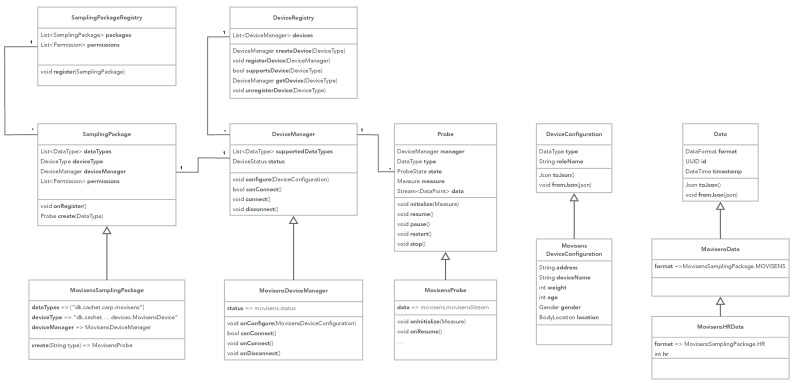
The static UML diagram of the sampling package pattern with the Movisens device as an example. *: Multiplicity.

**Figure 2 sensors-22-02813-f002:**
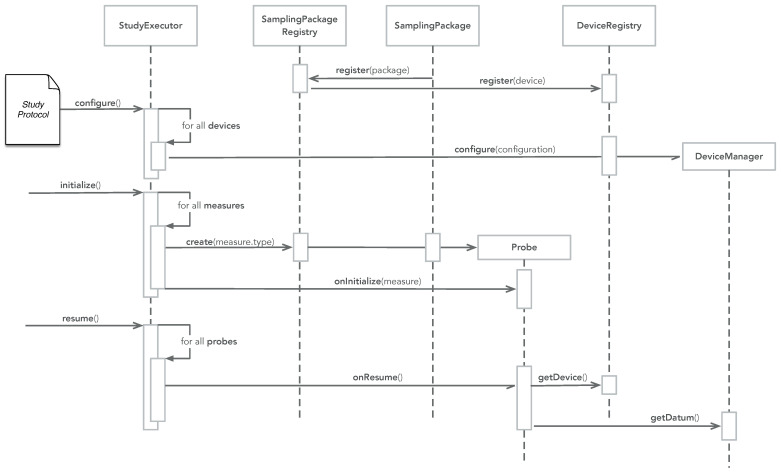
The UML interaction diagram of the sampling package pattern.

**Figure 3 sensors-22-02813-f003:**
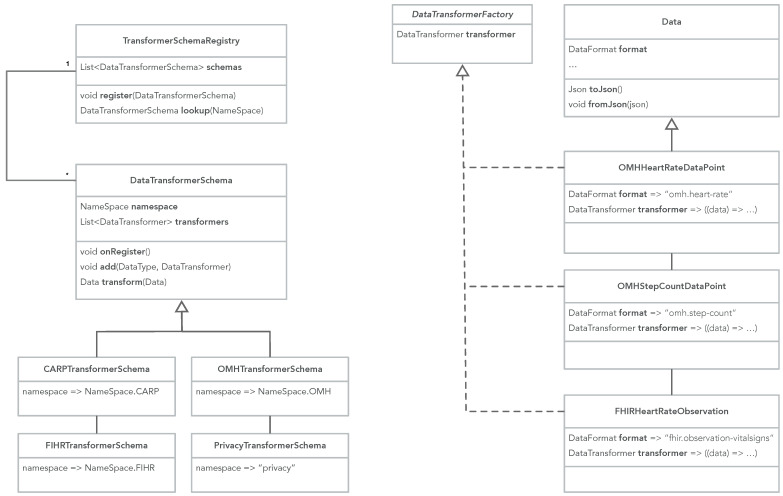
The UML class diagram of the ‘Data Transformer’ pattern with the HR data from the Movisens device being transformed into the OMH and FHIR data formats as an example. *: Multiplicity.

**Figure 4 sensors-22-02813-f004:**
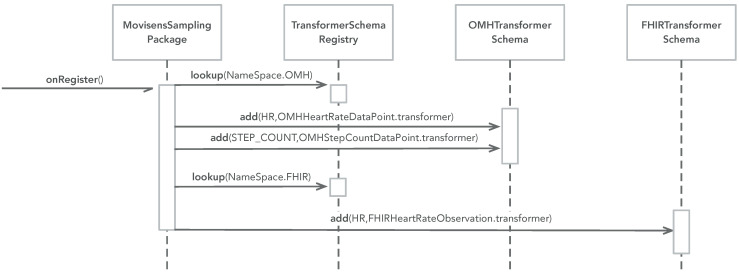
The UML interaction diagram of the setup phase of the data transformation pattern.

**Figure 5 sensors-22-02813-f005:**
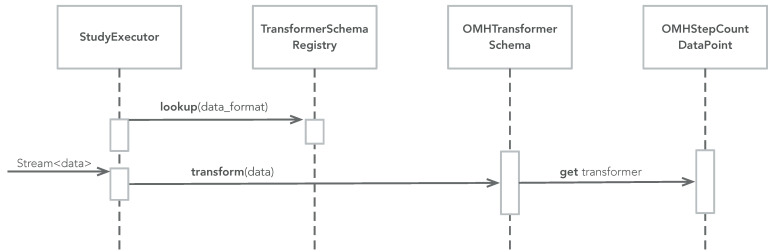
The UML interaction diagram of the transformation phase of the data transformation pattern.

**Figure 6 sensors-22-02813-f006:**
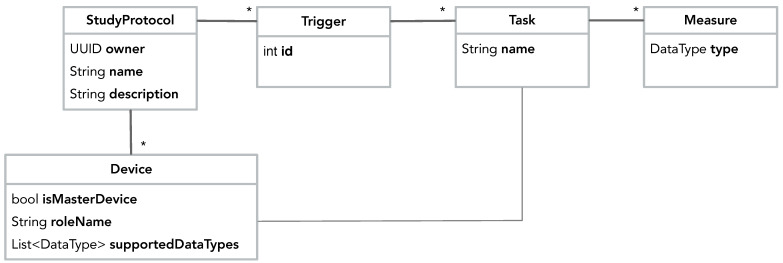
The UML class diagram of the StudyProtocol domain model. *: Multiplicity.

**Figure 7 sensors-22-02813-f007:**
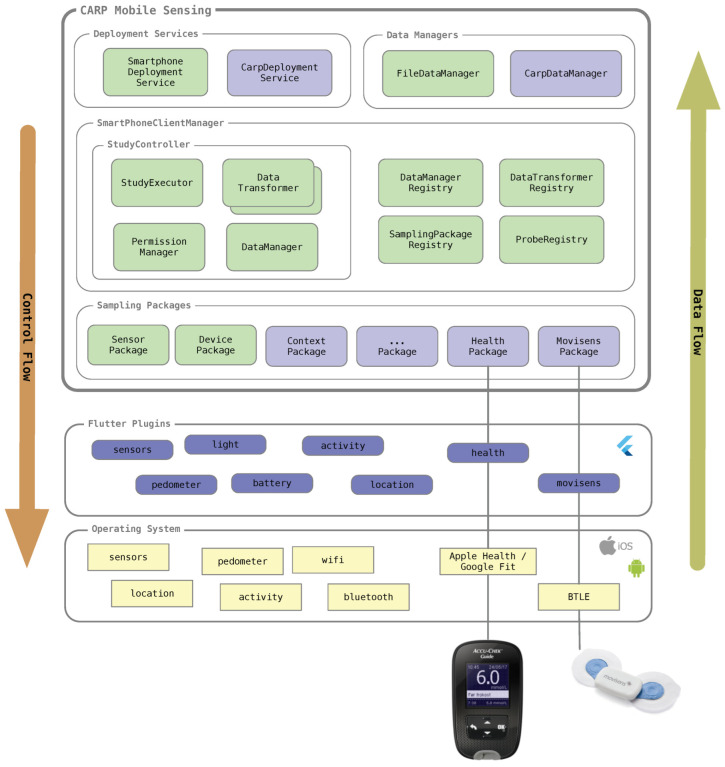
The overall software architecture and main components of the CARP Mobile Sensing (CAMS) framework. CAMS consists of three main layers; sampling packages, client manager, and services. Each sampling package uses one or more Flutter Plugins, which access processes, services, and data in the native OS or from external wearable devices (such as the AccuCheck Guide Blood Glucose Monitor (BGM) or the Movisens ECG devices). Sampling is controlled from the client manager and down to the OS whereas data flow from OS sensors, services, and wearable devices up towards the data managers.

**Figure 8 sensors-22-02813-f008:**
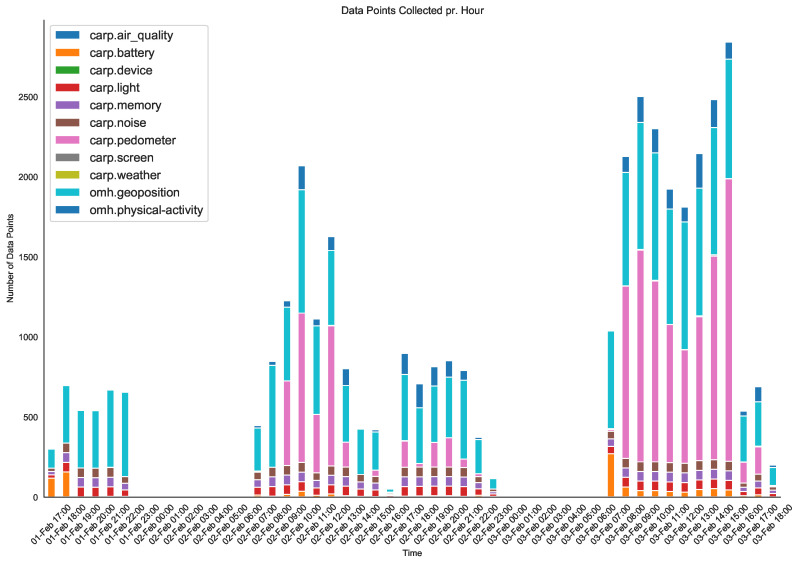
Data collection using the context, device, and sensor sampling packages. Hourly count of each data type in zulu (GMT) time.

**Figure 9 sensors-22-02813-f009:**
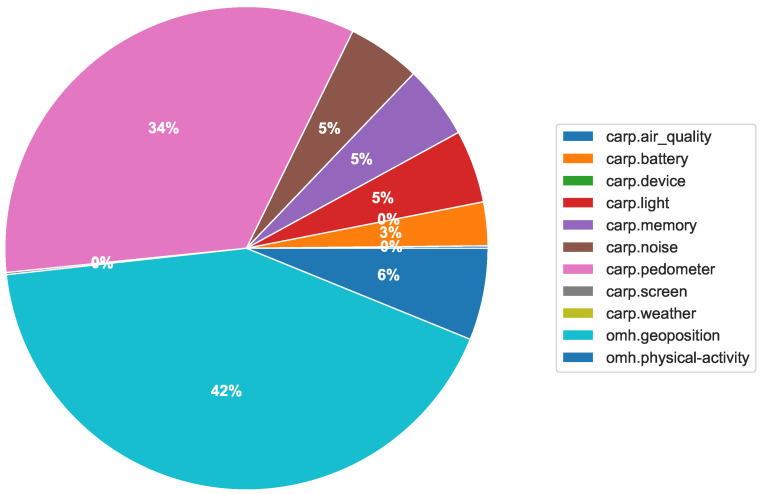
Distribution of collected data.

**Table 1 sensors-22-02813-t001:** Number of collected measures.

Measure	Count
carp.air_quality	61
carp.battery	1107
carp.device	1
carp.light	1850
carp.memory	1855
carp.noise	1857
carp.pedometer	12,846
carp.screen	55
carp.weather	1
omh.geoposition	15,973
omh.physical-activity	2332
Total	37,938

## Data Availability

The data and Jupyter Notebook with the Python scripts used in the reported study is available at https://github.com/cph-cachet/carp.analysis.sandbox/tree/master/projects/carp.mobile.sensing/transformation (accessed on 15 February 2022).

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
