# Peer review of "Software Architecture Patterns for Extending Sensing Capabilities and Data Formatting in Mobile Sensing"

_sensors, 2022, doi:10.3390/s22072813_

Round 1

Reviewer 1 Report

This paper presents an architecture for the development of cross-platform mobile applications for biomedical sensor data collection and processing (mHealth apps).

However, there are some issues in the paper that should be reviewed before its final publication. The first one has to do with the development framework chosen: Flutter. The article mentions the Dart programming language (line 137) and the Flutter framework (Figure 2) without first introducing them and without explaining what they are. Although later in the paper it is explained what Flutter and Dart are (line 388), this explanation should appear before the references I mentioned above. I also consider it necessary to extend this explanation about the framewrk and justify why it has been chosen over other cross-platform frameworks (React Native, Xamarin, etc.).

Another aspect that is not very clear in the proposed architecture is the issue of scalability. Since it is an topic that especially impacts on systems that collect and send a stream of sensor data, the issue of scalability is critical to prevent the saturation of the system when the number of sensors grows. This aspect should be considered in the paper.

Author Response

Thank you very much for the review, which has been very useful in improving the paper. There seemed to be some confusion as to what the paper was actually addressing and the main purpose of it. Therefore the paper has gone through a major revision, where most of the introduction and conclusion is re-written and the structure of the paper changed. 

Please let me address the individual issues separately below.

> This paper presents an architecture for the development of cross-platform mobile applications for biomedical sensor data collection and processing (mHealth apps).

This is partly true, but the main purpose of the paper is to present the two architecture design patterns, which enables a mobile sensing framework to be highly extensible. This main purpose and intented contribution is now much more highlighed by (i) rewriting the introduction and conclusion, and (ii) by re-organizing the paper so that these contributions are move up front.

> However, there are some issues in the paper that should be reviewed before its final publication. The first one has to do with the development framework chosen: Flutter. The article mentions the Dart programming language (line 137) and the Flutter framework (Figure 2) without first introducing them and without explaining what they are. Although later in the paper it is explained what Flutter and Dart are (line 388), this explanation should appear before the references I mentioned above. I also consider it necessary to extend this explanation about the framewrk and justify why it has been chosen over other cross-platform frameworks (React Native, Xamarin, etc.).

As said, the purpose of the paper is not to present or discuss CAMS and its selection of underlying technology. The purpose if to present the two design patterns, which can be implemented in any programming language and use React Native and/or Xamarin -- or native Android / iOS for that matter. 

However, the paper now contains some justification for choosen Flutter in CAMS in section 4.3

> Another aspect that is not very clear in the proposed architecture is the issue of scalability. Since it is an topic that especially impacts on systems that collect and send a stream of sensor data, the issue of scalability is critical to prevent the saturation of the system when the number of sensors grows. This aspect should be considered in the paper.

Again, scalability is not the focus of the paper (that is "extensibility"). However, the inclusion of the study in the end of the paper was to illustrate that CAMS and its architecture were able to support and collect data from a non-trivial study that sampled quite a lot of data over a long (2-3 day) period. The paper now have inlcuded Table 1 which illustrates the number of measures collected and hence reflects the scale of the study reported.

Reviewer 2 Report

The author of the paper entitled “ Software Architecture Patterns for Extending Sensing Capabilities and Data Formatting in Mobile Sensing ” presented a CARP Mobile Sensing (CAMS), which is a cross-platform (Android / iOS) software architecture  providing a reactive and unified programming model that emphasizes extensibility. The paper presents two software architecture patterns enabling dynamic extension of mobile sensing to incorporate new sensing capabilities, like collecting data from a wearable sensor, and handling real-time transformation of data into standardized medical data formats. The paper introduces the software architecture and programming model of CAMS, and shows how the framework uses the two software architecture patterns to add sampling support for an electrocardiography (ECG) device and support data transformation into the new Open mHealth (OMH) data format. The paper also presents data from a small study, demonstrating the robustness and feasibility of using CAMS for mobile sensing, including data transformation.

The paper is interesting, and the application is already added in a public platform. I think it is a nice and applicable study that can be accepted for publication after a minor revision, as follows:

- Some Listings and Figures can be further improved.

- More details about the implementation of the application can be added in Section (7-Study).

- English proofreading is also needed before publication.

Author Response

Thank you very much for the review, which has been very useful in improving the paper. There seemed to be some confusion as to what the paper was actually addressing and the main purpose of it. Therefore the paper has gone through a major revision, where most of the introduction and conclusion is re-written and the structure of the paper changed. 

Please let me address the individual issues separately below.

> The paper is interesting, and the application is already added in a public platform. I think it is a nice and applicable study that can be accepted for publication after a minor revision, as follows:

> Some Listings and Figures can be further improved.

I have updated the listings and figures. If there are any specific issues with, please let me know.

> More details about the implementation of the application can be added in Section (7-Study).

More details are now added, including Table 1. Also, the full study protocol is included in Appendix C.

> English proofreading is also needed before publication.

Yes. We have addressed some issues, but the final manuscript (once accepted for publication) will undergo proof-reading by a professional.

Reviewer 3 Report

The manuscript at hand presents a sensor framework that allows collecting sensor measurements from smart mobile devices (i.e., smartphones). The author shares insights into the overall development process, system architecture and design ideas. First insights from a study are presented.

First of all, i would like to thank the author for providing the opportunity to read and review this manuscript. The paper is very well written, clearly structured (i.e., there is a clearly visible red thread running through the paper) and easy to follow. Also, the motivation is very well illustrated. I really like the manuscript and the content presented. From a computer-science point-of-view it was a joy to read and follow the ideas of the author. 

When reading the manuscript, I found the following limitations, which should be considered when revising the work:
For me it is not clear, if the framework supports both major platforms (Android and iOS). Although the author describes, that CAMS is developed using Flutter (i.e., a cross-platform-development framework) no actual "proof" is shown. The study presented in Chapter 6, for example, only collects data from one person (severe limitation!) using an Android device (also limitation!). Table A2 shows which package (i.e., wifi, geolocation, ...) is available on which platform, but cannot be really tested. Would it be possible to show some "real life" screenshots of an realized application running on the Android and iOS Simulator (i.e., not in the browser) respectively? 

On Ionic (which is another cross-platform-development framework), support for "background jobs" on iOS are deprecated, because iOS does not allow for collecting / processing data any more, when the app is moved to the background (i.e., switch to another app). How does the CAMS framework deal with this issue? This may be a severe roadblock for the CAMS framework, when it can only collect data on iOS when in the foreground.

Furthermore, the manuscript does not provide a real "Related Work" discussion. The paper just lists some related projects from research and industry (i.e., Sensus, AWARE, ...) but does not discuss key features or benefits of these frameworks / products compared to the developed one. 

Finally, the manuscript lacks a critical discussion of the framework and the content presented in this manuscript. For example, limitations of the study (only one participant! only Android!) are not discussed at all.

Minor Issues:

  • [13] is missing some key information
  • Line 415: The reference to the table is not correctly resolved (i.e., it shows "??" instead of the correct number).
  • Line 493: The reference to the table is not correctly resolved (i.e., it shows "??" instead of the correct number).
  • Fig 8 & Fig 9: Colors are not the same across these 2 figures (i.e., "carp.light" is purpe in Fig 8 and red in Fig 9.). Also, the author state (Line 428), that there are 12 different types of data collected. Fig 9, however, only shows 11 (in the legend), and only 7 in the chart. Also, the title of Fig 9 ("Total Number of collected data") is misleading - it should be the "distribution of collected data", right?
  • There are 9 (out of 34) references citing the author himself.

Author Response

Thank you very much for the review, which has been very useful in improving the paper. There seemed to be some confusion as to what the paper was actually addressing and the main purpose of it. Therefore the paper has gone through a major revision, where most of the introduction and conclusion is re-written and the structure of the paper changed. 

Please let me address the individual issues separately below.

> The manuscript at hand presents a sensor framework that allows collecting sensor measurements from smart mobile devices (i.e., smartphones). The author shares insights into the overall development process, system architecture and design ideas. First insights from a study are presented.

The paper is actually not about the sensing framework (CAMS) as such, but more on the two design patterns ensuring extensibility and data transformation in sensing. Therefore, the paper is now rewritten and re-structured to make this intended contribution much more clear. 

> First of all, i would like to thank the author for providing the opportunity to read and review this manuscript. The paper is very well written, clearly structured (i.e., there is a clearly visible red thread running through the paper) and easy to follow. Also, the motivation is very well illustrated. I really like the manuscript and the content presented. From a computer-science point-of-view it was a joy to read and follow the ideas of the author. 

Thank you. I hope the current version still reads well and is even better structured according to its main message. The use of the design patterns style is exactly to keep it in a computer science / software architecture style of writing. 

> When reading the manuscript, I found the following limitations, which should be considered when revising the work:

> For me it is not clear, if the framework supports both major platforms (Android and iOS). Although the author describes, that CAMS is developed using Flutter (i.e., a cross-platform-development framework) no actual "proof" is shown. The study presented in Chapter 6, for example, only collects data from one person (severe limitation!) using an Android device (also limitation!). Table A2 shows which package (i.e., wifi, geolocation, ...) is available on which platform, but cannot be really tested. Would it be possible to show some "real life" screenshots of an realized application running on the Android and iOS Simulator (i.e., not in the browser) respectively? 

I fully understand these observations and comments. And even though these are relevant to a discussion/ presentation of CAMS, they are less relevant to the present paper. The cross-platform "proof" is in the application made using CAMS (these are now referenced in line 461-466 as references [27, 28, 29]). These studies have also collected data from many participants and represent "real life" applications. But since the purpose of the paper is NOT to present CAMS, these things are best not to add to this paper, but reference them (I think).

I hope that with the new introduction and structure to the paper that this becomes more clear.

> On Ionic (which is another cross-platform-development framework), support for "background jobs" on iOS are deprecated, because iOS does not allow for collecting / processing data any more, when the app is moved to the background (i.e., switch to another app). How does the CAMS framework deal with this issue? This may be a severe roadblock for the CAMS framework, when it can only collect data on iOS when in the foreground.

Again - this is topic for a discussion of CAMS (which is not the purpose of this paper). However, the issues of background sensing is presented in details both in the CAMS technical report [22] as well as in several blog post on the CARP home page (e.g., https://carp.cachet.dk/sampling-coverage-in-mobile-sensing-on-android-11/). 

The patterns and software architecture introduced in this paper does not necessitate background sensing - sensing can be done as an "active task" while the app is in the foreground. For example, collecting accelerometer data as part of a tremor analysis in Parkinsons' Disease.

> Furthermore, the manuscript does not provide a real "Related Work" discussion. The paper just lists some related projects from research and industry (i.e., Sensus, AWARE, ...) but does not discuss key features or benefits of these frameworks / products compared to the developed one. 

I was advised that in the SENSORS journal, there should not be a separate "related work" section. Instead it should be incorporated into the introduction which is now done in line 38-82. Moreover, there is a reference to a systematic literature review on mobile sensing frameworks, which we have presented earlier [11].

> Finally, the manuscript lacks a critical discussion of the framework and the content presented in this manuscript. For example, limitations of the study (only one participant! only Android!) are not discussed at all.

This is true. However, the purpose of the included study is not to run a large study involving multiple participants with multiple types of phones. The purpose of the study was to demonstrate the feasibility of the two design patterns, namly that a wide range of measures from different sampling packages could be configured in a study protocol and that the collected data could be correctly transformed from one format to another. 

Now more details on the study (including Table 1 and Appendix C) is included and the description is improved to reflect the intended purpose of the study.

-o-

Overall, I must say, however, that this review has been very useful since it provides valuable input of what should go into a paper that presents CAMS (which is in preparation). Hence, I will keep this review for this other paper.

-o-

> Minor Issues:

> [13] is missing some key information

> Line 415: The reference to the table is not correctly resolved (i.e., it shows "??" instead of the correct number).

> Line 493: The reference to the table is not correctly resolved (i.e., it shows "??" instead of the correct number).

> Fig 8 & Fig 9: Colors are not the same across these 2 figures (i.e., "carp.light" is purpe in Fig 8 and red in Fig 9.). Also, the author state (Line 428), that there are 12 different types of data collected. Fig 9, however, only shows 11 (in the legend), and only 7 in the chart. Also, the title of Fig 9 ("Total Number of collected data") is misleading - it should be the "distribution of collected data", right?

Thanks for spotting all of this. Fixed now.

> There are 9 (out of 34) references citing the author himself.

Yes - this happens as described above since we want to make references to other studies and apps build using the CAMS framework.

Round 2

Reviewer 3 Report

Dear Author,

thank you very much for writing back and discussing my review in detail. I can see your point and fully understand your arguments.

My issues were properly addressed by the author!

All the best